# Hyperlactatemia associated with elective tumor craniotomy: Protocol for an observational study of pathophysiology and clinical implications

**Alexandra Vassilieva**[1]⊙*, **Kirsten Møller**[1,2]⊙, **Jane Skjøth-Rasmussen**[3], **Martin Kryspin Sørensen**[1]⊙

1 Department of Neuroanaesthesiology, Rigshospitalet, Copenhagen, Denmark, 2 Department of Clinical Medicine, Faculty of Health Sciences, University of Copenhagen, Copenhagen, Denmark, 3 Department of Neurosurgery, Rigshospitalet, Copenhagen, Denmark

⊙ These authors contributed equally to this work.
* alexandra.vassilieva@regionh.dk

**Data Availability Statement:** No datasets were generated or analysed during the current study. All relevant data from this study will be made available

## Abstract

Hyperlactatemia occurs frequently after brain tumor surgery. Existing studies are scarce and predominantly retrospective, reporting inconsistent associations to new neurological deficits and prolonged hospital stay. Here we describe a protocol for a prospective observational study of hyperlactatemia during and after elective tumor craniotomy and the association with postoperative outcome, as well as selected pathophysiological aspects, and possible risk factors. We will include 450 brain tumor patients scheduled for elective craniotomy. Arterial blood samples for lactate and glucose measurement will be withdrawn hourly during surgery and until six hours postoperatively. To further explore the association of hyperlactatemia with perioperative insulin resistance, additional blood sampling measuring markers of insulin resistance will be done in 100 patients. Furthermore, in a subgroup of 20 patients, blood from a jugular bulb catheter will be drawn simultaneously with blood from the radial artery to measure the arterial to jugular venous concentration difference of lactate, in order to study the direction of cerebrovascular lactate flux. Functional clinical outcome will be determined by the modified Rankin Scale, length of stay and mortality at 30 days, 6 months, 1 year and 5 years. Clinical outcome will be compared between patients with and without hyperlactatemia. Multivariate logistic regression will be used to identify risk factors for hyperlactatemia. A statistical analysis plan will be publicized to support transparency and reproducibility. Results will be published in a peer-reviewed journal and presented at international conferences.

## Introduction

### Hyperlactatemia in brain tumor surgery: Clinical implications

Lactate metabolism in the human body is a complex, dynamic and incompletely understood process [1, 2]. In brain tumor surgery, hyperlactatemia occurs commonly with

upon study completion. Upon study completion, in compliance with Danish law and the Global Data Protection Regulation of the European Union, the collected data will not be made publicly available, but may be made available to interested parties upon reasonable request and preparation of a Data Sharing Agreement.

**Funding:** Alexandra Vassilieva has received funding from the following private foundation: 70.000 DDK from "Oberstinde Kirsten Jensa la Cours legat", 75.000 DDK from "Jens og Maren Thestrups Legat til kræftforskning" of and 25.000 DDk from "Agnethe Løvgreens Fond". None of these foundations provided a grant number or played a role in study design, preparation of protocol, data collection and analysis or decision to publish. The Department of Neuroanaesthesiology at Rigshospitalet, Denmark has received payment for performing quality control studies for Radiometer. Radiometer has had no influence on the use of this money and has not initiated or contributed to the design of this protocol. URL of each funder website: Oberstinde Kirsten Jensa la Cours legat: http://www.dasaim.dk/forskning/ansogninger-til-dasaims-fond-og-oberstinde-jensa-la-cours-legat/ Jens og Maren Thestrups Legat til kræftforskning: https://www.legatbogen.dk/jens-og-maren-thestrups-legat-til-krftforskning/stoetteomraade/6307 Agnethe Løvgreens Fond https://www.kvindeligelaeger.dk/om-legatet This does not alter our adherence to PLOS ONE policies on sharing data and materials.

**Competing interests:** There are no competing interests related to any patents, patent applications, or products in development or for market. The authors have declared that no competing interests exist.

studies reporting prevalences of 32–68% [3–8]. Yet, it is unclear whether hyperlactatemia is an independent risk factor for patients undergoing tumor craniotomy. In an older randomized trial in patients with severe burns, a reduction in type 1 lactic acidosis with dichloroacetate neither worsened nor improved hemodynamics or survival [9], suggesting that lactate is not toxic in itself, but rather a marker of disturbed homeostasis. However, it has also been suggested that lactate in patients with intracranial tumors may act as a substrate for tumor cells and promote tumor growth, and that hyperlactatemia thus constitutes an independent risk factor for cancer patients [10, 11]. Existing studies of hyperlactatemia in neurosurgery have not reached a consensus on whether lactate accumulation is related to clinical outcome. Thus, some have reported an association with longer hospital stay [5–8], a single study found a correlation with new neurological deficits [6], while others found no association with outcome [3, 4]. Available studies in this field are predominantly retrospective [3–7] with only one existing prospective study, which was terminated early based on an interim analysis indicating futility [8].

## Hyperlactatemia in brain tumor surgery: Pathophysiological implications

The cause of hyperlactatemia in brain tumor surgery is debated. Both tumor characteristics [10, 12–14], comorbidity [15], perioperative rhabdomyolysis [16], length of surgery [7], and type of anesthesia [17] have been suggested to play a part.

Lactate is well-known as a marker of global or regional ischemia, but as an additional consideration hyperlactatemia is also a marker of increased non-oxidative glycolysis and may be induced by hyperglycemia in healthy subjects [18]. Insulin resistance is a common side effect of glucocorticoid treatment and can also be induced in healthy subjects after short-term treatment [19]. As the majority of patients undergoing tumor craniotomy are treated with glucocorticoids either preoperatively (especially prednisolone to reduce edema and thus intracranial pressure), intraoperatively (especially dexamethasone, to reduce the perioperative stress response) or both, glucocorticoid-associated insulin resistance is an attractive explanation of hyperlactatemia after tumor craniotomy.

Lactate has previously been shown to act as a substrate for the brain in conditions with hyperlactatemia such as physical activity or lactate infusion [20]. It is therefore likely that perioperative hyperlactatemia associated with tumor craniotomy is also associated with increased cerebral lactate metabolism and, therefore, a net cerebrovascular lactate influx (i.e., a positive arterial to jugular venous concentration difference). This has never been addressed and would contrast with the usual finding of a slight net cerebrovascular lactate outflux from the brain to the bloodstream (i.e., a negative arterial to jugular venous concentration difference) at normal serum lactate.

The present research protocol describes a prospective study consisting of four sub-studies to further explore hyperlactatemia in tumor craniotomy patients. We intend to investigate whether hyperlactatemia is related to clinical outcome (sub-study I) and identify possible risk factors associated with hyperlactatemia (sub-study II). In subsets of patients, we aim to study whether hyperlactatemia in this context is associated with pre- and postoperative insulin resistance (sub-study III) as well as a net cerebrovascular influx of lactate (sub-study IV). We hypothesize that tumor size, preoperative glucocorticoid dose and the severity of hyperlactatemia is associated to functional clinical outcome, with preoperative glucocorticoid dose as the independent risk factor. Furthermore, we expect the severity of insulin resistance and hyperglycemia to be associated with the severity of hyperlactatemia, and that hyperlactatemia is associated with a net cerebrovascular influx of lactate.

## Methods and analysis

### Study design

This single-center, prospective observational cohort study will be conducted at the Department of Neurosurgery and Neuroanaesthesiology, Rigshospitalet, Copenhagen, Denmark.

### Study population

The Department of Neurosurgery and Neuroanaesthesiology, Rigshospitalet, performs an average of 20 elective craniotomies per week, most of these procedures are carried out on patients with a brain tumor. This study will include 450 elective tumor craniotomy patients.

**Inclusion criteria.** Patients who meet the following criteria will be included in the study: (1) patients with a full mental capacity who are $\geq$ 18 years old and scheduled for elective brain tumor craniotomy; (2) patients understanding written and spoken Danish.

**Exclusion criteria.** Patients who meet any of the following criteria will be excluded from the study: (1) scheduled for stereotactic biopsy; (2) scheduled for pituitary surgery; (3) lack of ability to provide written informed consent; (4) previously enrolled and scheduled for reoperation.

### Consent to participate

Patients selected and booked for elective tumor craniotomy will be approached during their preoperative visit and assessed for study eligibility. Potential participants will be provided with verbal and written information by the investigator and given an opportunity to ask questions about the in a private atmosphere with the ability to have a bystander. Potential candidates are allowed time to consider their willingness to participate. All participation is voluntary and requires written consent that can be withdrawn at any time.

### Risks, side effects and disadvantages for study participants

All participants will receive standard clinical treatment, supplemented by additional blood sampling from pre-existing intravascular catheters and two standardized neurological disability assessments using modified Rankin Scale (mRS). In a subgroup of 20 patients, jugular bulb catheterization will be performed after induction of general anesthesia. This ultrasound-guided procedure is associated with a minor risk of bleeding but is otherwise risk-free [21]. There is no increased risk of infection during blood sampling, as blood is drawn under aseptic conditions. Participants are covered by the insurance of Rigshospitalet and the Danish Patient Compensation Association.

### Sample size calculation

The sample size calculation is based on the study by Brallier et al (6). This study observed an incidence of new neurological deficits in 15% of craniotomy patients with S-lactate > 2.2 mmol/L, compared with 7% in craniotomy patients with S-lactate $\leq$ 2.2 mmol/L. A standardized difference of 0.26 was calculated. Assuming a power of 0.8 and a level of significance of 0.05 and using a graphical approximation [22] a sample size of 450 patients for the primary endpoint was reached.

### Data collection specifications

This study will acquire data from the patient medical charts, perioperative monitoring, blood analyses, two neurological disability assessments and the Danish Cause of Death Register (Fig 1).

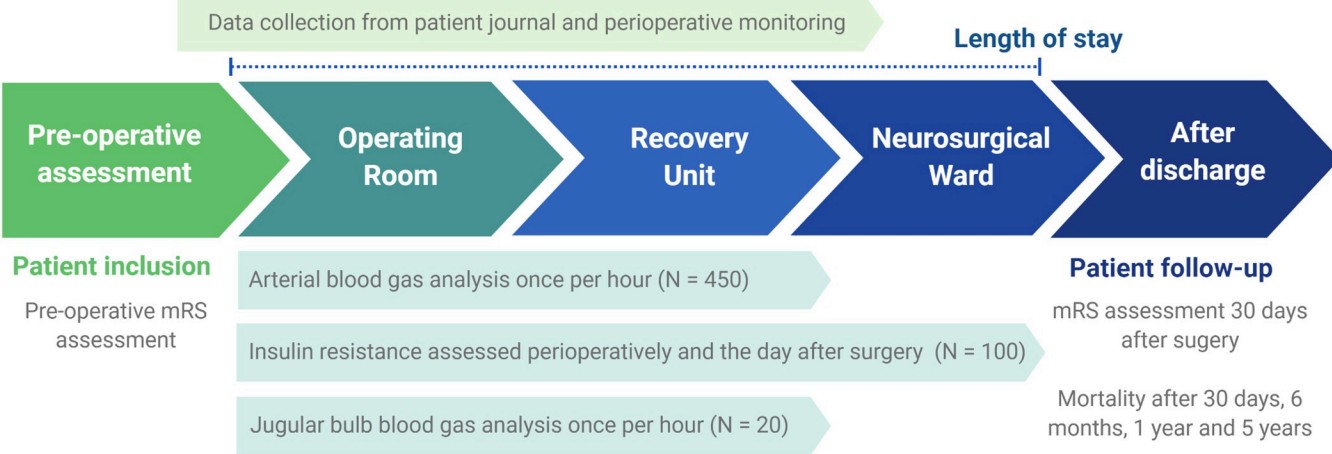

**Fig 1. Timeline for patient inclusion and data collection.** Patients are assessed for study eligibility during their preoperative visit. Preoperative mRS is assessed upon study inclusion. Hourly blood gas test from arterial cannula (N = 450) and jugular vein (N = 20) throughout surgery, starting at skin incision, and continuing for 6 postoperative hours in the Post Anesthesia Recovery Unit. Additional analyses of insulin resistance measured perioperatively and before food intake, the day after surgery, in a subgroup of patients (N = 100). Continuous data collection from perioperative monitoring and patient journal until discharge from hospital. Follow-up mRS assessment 30 days after surgery. Mortality data are obtained from the Danish Cause of Death Register after 30 days, 6 months, 1 year and 5 years.

**Preoperative data collection.** The following demographic information is retrieved from patient file: age, sex, BMI, preoperative neurologic disability assessed by mRS, comorbidity, tumor characteristics (including tumor location and volume from preoperative MRI), medication (including any glucocorticoid expressed as cortisol equivalents) and biochemistry.

**Intraoperative data collection.** Data collection from patient monitors (invasive and non-invasive blood pressure, saturation, heartrate, and ECG), medication, transfusions and events or complications during anesthesia / surgery.

Blood is drawn from arterial cannula (the cannula is inserted into the radial artery by an anesthesiologist shortly after induction of anesthesia, as part of the standardized perioperative monitoring) and subjected to analysis as follows:

- For all sub-studies, S-lactate, P-glucose, $SaO_2$ and $PaO_2$, using a point-of-care ABL800 FLEX blood gas analyzer (Radiometer) hourly from the time of skin incision to the end of surgery. (N = 450 patients)

- For sub-study III and IV, HbA1c, insulin, C-peptide, glucagon, and IL-6 immediately after induction and before treatment with intraoperative glucocorticoids. (N = 100 patients)

- For sub-study IV, ultrasound-guided retrograde catheterization (1-lumen, 16 G, 16 cm, Tele-flex/Arrow) of the internal jugular vein is inserted shortly after induction of general anesthesia. Analysis of arterial and jugular bulb levels of S-lactate, P-glucose and $PO_2$ are performed hourly as described above (N = 20 patients).

**Data collection in Post Anesthesia Recovery Unit.** Blood is drawn from arterial cannula immediately upon arrival to the Post Anesthesia Recovery Unit (before food intake) to analyze P-insulin, C-peptide, glucagon, and IL-6, after which patients are allowed food intake until 2 am (N = 100 patients). Hourly analysis of S-lactate, P-glucose, $SO_2$ and $PO_2$ is measured in blood from the arterial (N = 450 patients) and jugular bulb catheter (N = 20 patients). The arterial cannula and jugular bulb catheter are removed before discharge to the neurosurgical ward, typically after 6 hours observation. The presence and degree of nausea / vomiting as well

as pain, assessed by numeric or visual rating scale 0–10, are noted in the patient file at every hour. Time to mobilization in the Post Anesthesia Recovery Unit is noted as well.

**Data collection during follow-up.**   The morning after surgery, at 7.30 am (after fasting from 2.00 am) blood is drawn from a peripheral or central vein to analyze fasting P-glucose, P-insulin, C-peptide, glucagon, and IL-6 (N = 100 patients). Tumor type is classified by histopathological analysis of the intraoperative biopsy (classification level: glioma vs meningioma vs metastasis vs neurinoma vs cavernoma vs other). The discharge day and time is automatically recorded in the medical chart. The investigator estimates neurological disability using mRS at follow-up assessment or over telephone 30 days after surgery. Mortality after 30 days, 6 months, 1 and 5 years is obtained from the Danish Cause of Death Register.

## Data management

Patient information is collected and stored by the investigators in accordance with the General Data Protection Regulation with consent and supervision from the Danish Data Protection Agency.

Data are kept confidential in Research Electronic Data Capture, a password-protected software that stores data in case report forms with restricted and secured access. Case report forms are only accessible to investigators and coinvestigators. Exported files are stored on an encrypted personal computer with limited access.

## Definitions and calculations

Hyperlactatemia is defined as the measurement of at least one value of S-lactate > 2.2 mmol / L during the observation period starting at surgical incision and ending after 6-hours of observation in the Post Anesthesia Recovery Unit.

Peak lactate is defined as the highest value of S-lactate measured during the observation period.

Mean S-lactate is calculated as the area under the curve for S-lactate over time during the observation period, divided by the duration in hours. This is calculated to estimate the "total" lactate load regardless of whether the triggers occur pre-, intra- or postoperatively.

Intra- / postoperative lactate excess is calculated as mean S-lactate minus S-lactate at surgical incision. This is calculated to estimate the "pure" perioperative contribution to the lactate load.

Arterio-venous concentration difference (a-jD) for lactate, glucose, and oxygen: the concentration in jugular bulb blood minus the concentration in arterial blood.

## Study endpoints

**Primary endpoints.**   Sub-study I: Change in mRS from preoperative assessment to 30 days after surgery, in patients with and without hyperlactatemia.

Sub-study II: Prevalence of hyperlactatemia (defined as at least one measurement of S-lactate > 2.2 mmol / L).

Sub-study III: Association between preoperative and/or intraoperative glucocorticoid treatment, HbA1c and pre-/ postoperative insulin resistance (measured by HOMA-IR), adjusted for hours after surgery.

Sub-study IV: Perioperative change in a-jD (lactate) corrected for change in a-jD ($O_2$) from first to last measurement.

**Secondary endpoints.**   Sub-study I:

- Length of hospital stay, from day of surgery to discharge, in days.

- Mortality at 30 days, 6 months, 1 year and 5 years (6 months, 1 year and 5 years mortality will not be included in the initial data analysis and publication).

  Sub-study II:

- Association between pre- / intraoperative risk factors (specified in exploratory endpoints) and hyperlactatemia.

  **Exploratory endpoints.**

- Tumor type, location and estimated volume on preoperative MRI scan.

- BMI and comorbidity (Charlson Comorbidity Index), with focus on insulin dependent or non-dependent diabetes mellitus type 1 or 2 and liver / kidney disease.

- Neurological deficits at admission and at discharge. The deficits are assessed by the admitting and discharging clinician through standard neurological examination (mental status, cranial nerves, motor system, reflexes, sensation, and cerebellar function).

- Preoperative chemotherapy, radiation, or immunotherapy.

- Dose and duration of pre-, intra-, and post-operative treatment with glucocorticoids (expressed as cortisol equivalents in mg).

- Surgery duration (minutes) and surgery events (such as other procedures during the surgery and complications, using Landriel Ibañez classification).

- Position during surgery

- Complete or incomplete total resection of tumor, as indicated by the neurosurgeon in the operative report.

- Type of anesthesia: total intravenous anesthesia, inhalation anesthesia, awake craniotomy.

- Intra- and post-operative hypotension (defined as MAP < 60 for > 15 min or MAP < 30% of preoperative resting MAP for > 15 min) and treatment with vasopressors (ephedrine, noradrenaline, methoxyarene) in mg for total surgery duration.

- Perioperative fluid therapy volume and type, as well as total urine output in mL / hour during surgery and in the Post Anesthesia Recovery Unit.

- Estimated blood loss and blood transfusion volume.

- Perioperative blood gas parameters: pH, glucose, $PaO_2$, $PaCO_2$, Base Excess, Hb, $HCO_3$, Na, K, Cl.

- Postoperative nausea, vomiting and pain. Pain assessed by numeric or visual rating scale 0–10.

- Mobilization time to upright sitting position in the Post Anesthesia Recovery Unit, measured in hours.

## Statistical analysis

Parametric statistics will be used, if data cannot be assumed to follow a normal distribution. P <0.05 is considered statistically significant. Patient characteristics and continuous variables are presented as median values (interquartile range). In the outcome analysis, lactate is analyzed both as continuous (S-lactate area under the curve, total and intraoperative lactate load) and binary variable (S-lactate above and below 2.2 mmol / L).

Linear and multivariate logistic regression will be used for continuous and binary variables respectively. RStudio or comparable professional statistical program will be used. The statistical analysis plan will be made publicly available.

## Ethics approval and dissemination of results

The study adheres to Good Clinical Practice guidelines. The protocol has been approved by the Committee on Health Research Ethics of the Capital Region of Denmark (approval date: 21.07.2020, identifier: H-20011650) and by the Danish Data Protection Agency (approval date: 18.05.2020, identifier: P-2020-566). The study is registered on clinicaltrials.gov (identifier: NCT04410315). Amendments to the protocol will be communicated to the relevant authorities.

The results of the study (positive, negative, or inconclusive) will be published in a peer reviewed journal. The results will also be presented in relevant conferences. Reporting will be guided by "Strengthening the Reporting of Observational Studies in Epidemiology Statement".

## Trial status

The study has been recruiting patients since the 1st of August 2020. We estimate a trial recruitment period of 18–20 months.

## Bias and confounders

Selection bias is almost always present in studies with voluntary participation. Factors such as socioeconomic status, psychological resilience, severity of illness and number of comorbidities may influence the patient's readiness to participate. The published data will therefore include the number of non-participants.

Furthermore, patients are included consecutively, in the order in which the studies are mentioned.

Since investigators are susceptible to observational bias, a flowchart and standardized questioning will be used for mRS assessment.

We expect minimal loss to follow-up regarding mortality, as this information will be accessed through the Danish Cause of Death Register. Because many patients may die during the follow-up, survivorship bias is an important type of bias and will be corrected for as appropriate.

The measurement error of S-lactate is expected to be minimal, since all blood samples are analyzed with the same type of equipment.

Major factors influencing lactate production and metabolism are investigated in the study as exploratory endpoints. Continuous variables, such as vital data, are measured on the same type of monitor for all patients. To avoid misclassification, tumors will be classified using the 10th version of International Classification of Disease and the World Health Organization's classification of tumors, based on the histopathological conclusion from the intraoperative biopsy. The standardized case report form will be publicized as a supplement to study results. Missing data and number of patients lost to follow-up will be reported for each group (normo-lactatemia and hyperlactatemia) to address any distortion of study results.

We will avoid publication bias by attempting to publish the study irrespective of its conclusions.

## Discussion

Even though hyperlactatemia is common in tumor craniotomy patients, little is known of the underlying pathophysiology and the clinical effect of this phenomenon. The results of this

study will provide new knowledge about the incidence and possible risk factors for hyperlactatemia in patients undergoing tumor craniotomy, as well as the potential association with functional outcome. It will also explore the role of the surgical stress response and glucocorticoid-associated insulin resistance for the development of hyperlactatemia and provide an indication of the pathophysiological significance for the brain by measuring the net cerebrovascular flux of lactate.

Previous studies on this subject [3–7] have been retrospective and may therefore have been susceptible to selection bias. With a prospective study we hope to avoid such bias by sampling data according to a prespecified protocol. There are, however, possible contributing factors to hyperlactatemia that we have not accounted for in our design. Thus, perioperative rhabdomyolysis may increase S-lactate [23]. In neurosurgery, rhabdomyolysis has been reported amongst patients who are placed in the lateral position, and for those who undergo surgery lasting more than 5 hours [16]. Myoglobin, creatine kinase, lactate dehydrogenase and creatinine are measured in our department only if there is clinical suspicion of rhabdomyolysis or kidney failure. We would most likely detect severe rhabdomyolysis, as we routinely monitor patient mobility, urine output and electrolyte levels. As part of our study design, we record perioperative positioning, duration of surgery, and BMI as possible contributing risk factors for hyperlactatemia.

Additionally, the type and volume of intravenous fluid administration may decrease or increase lactate levels by dilution or exogenous contribution, respectively. The intravenous fluid of choice in our department is isotonic saline; selected patients may receive hypertonic saline or Ringer lactate on specific indications. We do not use mannitol. Intravenous fluid type and volume in each perioperative phase will be reported in the final manuscript.

Another limitation of our study is the time period for lactate measurements. Due to the organization of perioperative care, we cannot measure S-lactate for longer that 6 hours postoperatively and therefore cannot differentiate between patients with late, prolonged, or short-lived hyperlactatemia.

An observational study design makes it possible to address multiple possible risk factors at once, which is a necessary approach when investigating a multifactorial phenomenon [24], such as hyperlactatemia. Many risk factors, however, may interact with each other. An example of this is the interaction between tumor volume and glucocorticoid dose. Tumor cells produce lactate, a process known as the Warburg effect [25]. A large tumor load may therefore contribute to elevated lactate levels [26]. Furthermore, voluminous and infiltrating brain tumors bring about peritumoral edema with increased intracranial pressure, a condition that is typically remedied with high-dose glucocorticoids, a drug class which may raise lactate through insulin resistance with hyperglycemia [27]. Tumor volume and glucocorticoids are thus interacting risk factors that contribute to lactate production through different mechanisms.

Moreover, lactate production by the tumor itself may be notably higher in patients with brain metastasis and extracranial primary tumors. We expect that these patients will exhibit higher initial lactate levels, but an unchanged "pure" perioperative contribution to the lactate load (see "Definitions and calculations"). Conversely, we expect that initial lactate levels are lower in patients who have received preoperative chemotherapy, radiation or immunotherapy, due to a reduction in tumor burden.

Brain tumors are very diverse, but it is unknown whether the lactate profile and time course differs by tumor type. We have chosen to include patients with meningiomas in our study, as they are classified as brain tumors by the American Association of Neurological Surgeons, and because previous studies have reported on the occurrence of hyperlactatemia even in patients with this tumor type. In the future, it also will be interesting to assess the effect of craniotomy on hyperlactatemia in patients with other surgical (e.g. neurovascular) indications.

While observational studies may be helpful at recognizing associations between variables, they are not designed to study causality. This study may thus help uncover associations between hyperlactatemia and potential risk factors, which subsequently will need further scrutiny to suggest potential causality. We therefore hypothesize that glucocorticoid-induced insulin resistance is a major risk factor of hyperlactatemia, i.e. that there is an association between glucocorticoid doses, markers of insulin resistance, and arterial lactate levels. If this hypothesis appears to be corroborated and if hyperlactatemia appears to be an independent risk factor of functional outcome, a randomized trial of two different doses of perioperative glucocorticoid may be planned.

## Supporting information

**S1 File. Strobe checklist.** Strobe checklist for "Hyperlactatemia associated with elective tumor craniotomy: protocol for an observational study of pathophysiology and clinical implications". (DOC)

## Acknowledgments

We thank Niels Risør Hammer (Department of Neuroanaesthesiology, Rigshospitalet) and Morten Ziebell (Department of Neurosurgery, Rigshospitalet) for supporting this research project with department resources. We also wish to thank Signe Tellerup Nielsen (Department of Anaesthesia, Bispebjerg Hospital) and Rikke Krogh-Madsen (Center for Physical Activity Research, Rigshospitalet) for insightful discussions concerning insulin resistance in planning of sub-study III.

## Author Contributions

**Conceptualization:** Kirsten Møller, Martin Kryspin Sørensen.

**Funding acquisition:** Alexandra Vassilieva, Kirsten Møller, Martin Kryspin Sørensen.

**Investigation:** Alexandra Vassilieva.

**Methodology:** Alexandra Vassilieva, Kirsten Møller, Martin Kryspin Sørensen.

**Project administration:** Alexandra Vassilieva, Jane Skjøth-Rasmussen, Martin Kryspin Sørensen.

**Supervision:** Kirsten Møller, Martin Kryspin Sørensen.

**Writing – original draft:** Alexandra Vassilieva.

**Writing – review & editing:** Kirsten Møller, Jane Skjøth-Rasmussen, Martin Kryspin Sørensen.

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
