## [Decision Letter · Decision Letter 0]

10 Mar 2022

PONE-D-21-37350Hyperlactatemia associated with elective tumor craniotomy: protocol for an observational study of pathophysiology and clinical implications.PLOS ONE

Dear Dr. Vassilieva,

Thank you for submitting your manuscript to PLOS ONE. After careful consideration, we feel that it has merit but does not fully meet PLOS ONE’s publication criteria as it currently stands. Therefore, we invite you to submit a revised version of the manuscript that addresses the points raised during the review process.

ACADEMIC EDITOR: Reviewers found potential in your manuscript, however it needs an extensione revision before considering accpentace. Please follow the attached comments.

We look forward to receiving your revised manuscript.

Kind regards,

Alfio Spina, M.D.

Academic Editor

PLOS ONE

Journal Requirements:

Reviewers' comments:

Reviewer #1: Thank you for submitting the study protocol.

This is a quite an endeavour to be undertaken.

1. What is the expected time frame for completion? With these difficult covid times, still possible to undertake such volume load?

2. How does lactate relate to rhabdomyolysis?

3. Would patient positioning affect lactate level as well? Patient mobility pre and post op?

4. A patient with know primary cancer would still be included in the study? Their preop lactate level may be more elevated compared to the standard?

5. Does the lactate level correlate with the preoperative dose and time duration of dexamethasone treatment?

6. Whether or not the patient underwent previous chemotherapy treatment or immunotherapy treatment affect lactate levels?

7. Why are samples collected until 6 hrs post op? why not till they ambulate or 12 hrs?

8. Does the type of IV fluid given pre, intra and post op vary? Normal saline vs ½ NS vs Ringer lactate?

9. Are patients given any 20% mannitol pre or intra or post op?

10. How about hypertonic saline solution?

11. What is the expected loss of follow up rate? Especially if a patient is diagnosed with a glioblastoma, survival till 5 year is rare.

12. Are patient not allowed to eat until the next morning to asses their fasting glucose level? Even if they have no nausea and want to eat?

13. How does doing an intraoperative frozen section help in this study?

14. All patients are observed 6 hours in the PACU…so no discharge to the ICU or intermediate zone before?

Reviewer #2: Given the fact that the study is recruiting patients since 2020 discussion of the study protocol at this time appears futile. The underlying assumption of the study is an association of peri/postoperative hyperlactatemia in patients undergoing elective brain Tumor surgery. 450 patients are planned to be included of which 100 patients will receive additional testing for Insulin resistance and further 20 patients will have the arterial to jugular venous concentration difference tested.

My Major concern focusses on Patient selection. Adult patients scheduled for elective brain tumor surgery are eligible for study inclusion. However, brain tumor is not a useful classification, as it is not a coherent clinical or pathological entity. The authors determine tumor size and location. I could not find more details concerning tumor classification. However there is a relevant difference between the different types of intracranial tumors, not only in location and size. First, of all 37% of intracranial tumors are benign meningiomas. These tumors cannot be classified as “brain tumors” per se. In contrast, the largest group of intra-axial tumors are the the astrocytic gliomas. Within this group are highly malign tumors as well as rather benign entities. A third relevant group are brain metastasis with a wide variety of tumor biology. The authors mention the Warburg-effect as a relevant contributor to lactate levels. The existence of this effect in malign brain tumors such as the glioblastoma and brain metastasis has been described. However, the relevance of this specific tumor metabolism in meningiomas is not well understood. For a prospective study I would expect a more detailed definition of included patients concerning the tumor diagnosis. If hyperlactatemia is not induced by the tumor but by the procedure (that is craniotomy and intracranial surgery), other indications for surgery should be included.

Another aspect of patient selection, which is not clarified in the protocol, is the process of selecting which patient receives an additional investigation either insulin resistance or a jugular bulb catheter for extended blood probes. Are these patients randomly chosen? If so, how is this done?

There are other minor imprecise statements within this study protocol. For patients who are chosen or qualify (?) for additional insulin resistance analysis, blood samples are taken “the morning after surgery”. I would assume a more precise definition of time (e. g. hours after end of surgery) is necessary to guarantee for comparability of data.

There are many confounding factors for hyperlactatemia, for instance post- or intraoperative complications and preoperative comorbidities. How are these identified and registered? Are common scoring systems used? (e. g. Charlson Comorbidity Index, Clavien-Dindo classification system).

How is neurological status recorded? The authors state that status is recorded “through standard neurological examination”. To make data comparable more details of this standard examination would be necessary.

---

## [Author Response · Author response to Decision Letter 0]

3 Jun 2022

Dear Dr Spina and reviewers,

Thank you for your thorough work and insightful comments to our manuscript. We have revised it according to your suggestions and believe that it has improved considerably during this process. 

Reviewer #1: Thank you for submitting the study protocol.

This is a quite an endeavour to be undertaken.

Thank you for your kind comment. 

1. What is the expected time frame for completion? With these difficult covid times, still possible to undertake such volume load?

Many research projects have indeed been delayed by the pandemic. Luckily, cancer surgery in Denmark has been minimally affected, and the project has therefore been running smoothly. Currently, all 450 patients have been included in the study; the last patient was included in February, 2022. We have, however, not completed data sampling.

2. How does lactate relate to rhabdomyolysis?

Damaged and ischemic skeletal muscle cells may produce lactate, and rhabdomyolysis may thus be associated with increasing S-lactate levels. In neurosurgery, rhabdomyolysis has been reported in patients who are placed in the lateral position, and with durations of surgery exceeding 5 hours. However, rhabdomyolysis could be an underdiagnosed condition, especially in patients with high BMI. 

We have addressed this in the manuscript on page 3, line 59 and page 15-16, line 307-315.

3. Would patient positioning affect lactate level as well? Patient mobility pre and post op?

Thank you for this suggestion. We have added position during surgery as an exploratory endpoint in the study (page 12, line 237). We evaluate patient mobilization in the Post Anesthesia Recovery Unit as a separate exploratory endpoint. 

4. A patient with known primary cancer would still be included in the study? Their preop lactate level may be more elevated compared to the standard?

We agree with the reviewer that this is an important issue. We include patients with brain metastasis in the study. As tumor cells produce lactate, patients with greater tumor load, such as patients with both a primary tumor and metastases, may have higher S-lactate levels. First measurement of lactate might reflect this tendency, with higher initial S-lactate levels amongst patients who undergo tumor craniotomy due to a metastasis. We will therefor calculate the “pure” perioperative contribution of lactate load (e.g. mean S-lactate minus S-lactate at surgical incision, see “Definitions and calculations”). 

We have clarified this on page 16-17, line 334-338.

5. Does the lactate level correlate with the preoperative dose and time duration of dexamethasone treatment?

We thank the reviewer for raising this point, as this is one the most important questions of the study and one of our main hypotheses (please see page 17, line 347-350). To make this clearer to the reader, we have elaborated on this in the section on exploratory outcomes (page 12, line 233-234)

Dexamethasone may elevate lactate perioperatively. We have added changes and a reference (page 16, line 329-332) to substantiate this statement.

6. Whether or not the patient underwent previous chemotherapy treatment or immunotherapy treatment affect lactate levels?

In theory, preoperative chemotherapy, radiation, or immunotherapy should reduce the tumor load and therefore decrease any tumor-induced lactate elevation. Preoperative chemotherapy is one of our exploratory outcomes. We have now added preoperative radiation and immunotherapy as exploratory outcomes (page 12, line 232) and elaborated on the possible effect of such preoperative treatment on lactate levels (page 17, line 337-338).

7. Why are samples collected until 6 hrs post op? why not till they ambulate or 12 hrs?

We agree that it would have been interesting to measure lactate for a longer period. This design was chosen due to the structure and organization of our department. With a few exceptions, patient undergoing tumor craniotomy are observed for 6 hours in the Post Anesthesia Recovery Unit and are thereafter moved to the Neurosurgery Ward. Arterial puncture is associated with discomfort and complications such as hematoma, occlusion and pseudoaneurysm. We therefore designed the study with hourly blood samples from an arterial cannula that is placed preoperatively as part of the standard perioperative monitoring. Upon discharge from the Recovery Unit, the arterial cannula is removed. It has therefore not been possible for us to continue with our hourly measurements of lactate after transfer to the Neurosurgery Ward. We elaborate on this on page 9, line 170 and page 16, line 321-323.

8. Does the type of IV fluid given pre, intra and post op vary? Normal saline vs ½ NS vs Ringer lactate?

As an exploratory outcome for hyperlactatemia, we wish to study IV fluid volume and type during surgery and in the Post Anesthesia Recovery unit (clarified on page 12, line 244). The standard in our department is IV fluid treatment with isotonic saline, although some patients do receive Ringer lactate upon clinical indication (we have added a description of this on page 16, line 317-320). 

9. Are patients given any 20% mannitol pre or intra or post op?

We do not use mannitol in our department. This has been clarified on page 16, line 319. 

10. How about hypertonic saline solution?

This will be registered together with other IV treatment type and volume (page 16, line 318-320).

11. What is the expected loss of follow up rate? Especially if a patient is diagnosed with a glioblastoma, survival till 5 year is rare. 

This is a very important concern as many patients probably will die during the follow up. We have addressed this on page 14, line 283-285.

12. Are patient not allowed to eat until the next morning to asses their fasting glucose level? Even if they have no nausea and want to eat?

Thank you for the opportunity to clarify this point. Patients are allowed to eat and drink from their arrival in the Post Anesthesia Recovery Unit and in the Neurosurgery Ward. The day after surgery, they will be fasting from 2.00 am until blood sampling at 7.30 am. We have added this clarification to the manuscript on page 9, line 167-168 and page 9, line 176.

13. How does doing an intraoperative frozen section help in this study?

As per clinical routine, we take an intraoperative biopsy of the tumor, which is then processed at the Department of Pathology. In our study, we use this histopathological analysis to classify tumor type (see page 9, line 177-178), in order to analyze potentially different profiles of hyperlactatemia between different tumor types. 

14. All patients are observed 6 hours in the PACU…so no discharge to the ICU or intermediate zone before?

Yes, that is correct. Unless patients develop special requirement for intensive care, they are discharged to the Neurosurgery Ward. The Post Anesthesia Recovery Unit is subdivision of the Department of Neurointensive Care.

Reviewer #2:

Given the fact that the study is recruiting patients since 2020 discussion of the study protocol at this time appears futile. The underlying assumption of the study is an association of peri/postoperative hyperlactatemia in patients undergoing elective brain Tumor surgery. 450 patients are planned to be included of which 100 patients will receive additional testing for Insulin resistance and further 20 patients will have the arterial to jugular venous concentration difference tested.

My Major concern focusses on Patient selection. Adult patients scheduled for elective brain tumor surgery are eligible for study inclusion. However, brain tumor is not a useful classification, as it is not a coherent clinical or pathological entity. The authors determine tumor size and location. I could not find more details concerning tumor classification. However there is a relevant difference between the different types of intracranial tumors, not only in location and size. First, of all 37% of intracranial tumors are benign meningiomas. These tumors cannot be classified as “brain tumors” per se. In contrast, the largest group of intra-axial tumors are the the astrocytic gliomas. Within this group are highly malign tumors as well as rather benign entities. A third relevant group are brain metastasis with a wide variety of tumor biology. The authors mention the Warburg-effect as a relevant contributor to lactate levels. The existence of this effect in malign brain tumors such as the glioblastoma and brain metastasis has been described. However, the relevance of this specific tumor metabolism in meningiomas is not well understood. For a prospective study I would expect a more detailed definition of included patients concerning the tumor diagnosis. If hyperlactatemia is not induced by the tumor but by the procedure (that is craniotomy and intracranial surgery), other indications for surgery should be included.

We thank the review for these salient remarks. We fully agree that intracranial tumors are highly diverse regarding type, size, and location; even so, hyperlactatemia has been observed perioperatively across these different characteristics. That said, we fully agree with the reviewer that this is an important opportunity to analyze potentially different lactate profiles by these different tumor characteristics. Classification of tumor type will be based on histopathological finding of the Department of Pathology from the tissue collected intraoperatively (please see page 9, line 177-179, where we have added a level of classification to avoid p hacking), as we believe that this is the most correct way to differentiate between tumor types. We have added a clarification on page 14-15, line 291-292.

We also agree that benign meningiomas are not “brain tumors” per se, even though some classification systems (e.g. American Association of Neurological Surgeons) define it as such. As we initially set out to study hyperlactatemia across tumor types, aiming to analyze the profile of lactate by tumor type, we have included meningiomas of all grades in our study. 

It would also be interesting to study hyperlactatemia in regard to procedure. We plan on doing a pilot study with the same design on patients undergoing craniotomy to treat unruptured intracranial aneurisms. 

We have added page 17, line 339-344 to elucidate these important points.

Another aspect of patient selection, which is not clarified in the protocol, is the process of selecting which patient receives an additional investigation either insulin resistance or a jugular bulb catheter for extended blood probes. Are these patients randomly chosen? If so, how is this done?

This is a highly relevant remark and we thank the reviewer for making it. All patients are included consecutively, in the order in which the studies are mentioned. This consecutive order also applies for investigation and patient data collection. We believe that patients represent a random sample of the population of patients with intracranial tumors admitted to our hospital, covering the catchment area of eastern Denmark, regardless of tumor type, size, or location. We have added a clarification of this on page 14, line 280. 

There are other minor imprecise statements within this study protocol. For patients who are chosen or qualify (?) for additional insulin resistance analysis, blood samples are taken “the morning after surgery”. I would assume a more precise definition of time (e. g. hours after end of surgery) is necessary to guarantee for comparability of data.

The morning sampling rounds in our department occur at 7-8 am. We have added a more precise time for blood samples taken on the morning after surgery on page 9, line 176. When comparing the analysis, it will be important to adjust the measurements for hours after surgery as surgery starting time varies between patients (changes added on page 11, line 212). 

There are many confounding factors for hyperlactatemia, for instance post- or intraoperative complications and preoperative comorbidities. How are these identified and registered? Are common scoring systems used? (e. g. Charlson Comorbidity Index, Clavien-Dindo classification system).

This is an excellent point and suggestion. We have added these specifications on page 11, line 227 and page 12, line 236 Preoperative morbidity will be registered by the Charlson Comorbidity Index and the severity of postoperative complications will be classified by the Landriel Ibañez classification system, a four-grade scale based on the therapy used to treat the neurosurgical complications, similarly to the Clavien-Dindo classification system.

How is neurological status recorded? The authors state that status is recorded “through standard neurological examination”. To make data comparable more details of this standard examination would be necessary.

Patients are examined at admission and discharge and any neurological deficits will be recorded in the patient journal. We focus on new neurological deficits, which are always noted in the patient journal.

We have now added the relevant changes to our manuscript based on the excellent suggestion from the reviewers. Thank you again for taking the time to make our work more thorough and well-considered. 

Kind regards, 

Alexandra Vassilieva, Kirsten Møller, Jane Skjøth-Rasmussen and Martin Kryspin Sørensen

---

## [Editor Report · Decision Letter 1]

6 Jul 2022

Hyperlactatemia associated with elective tumor craniotomy: protocol for an observational study of pathophysiology and clinical implications.

PONE-D-21-37350R1

Dear Dr. Vassilieva,

We’re pleased to inform you that your manuscript has been judged scientifically suitable for publication and will be formally accepted for publication once it meets all outstanding technical requirements.

Kind regards,

Alfio Spina, M.D.

Academic Editor

PLOS ONE
---

## [Editor Report · Acceptance letter]

12 Jul 2022

PONE-D-21-37350R1 

Hyperlactatemia associated with elective tumor craniotomy: protocol for an observational study of pathophysiology and clinical implications. 

Dear Dr. Vassilieva:

I'm pleased to inform you that your manuscript has been deemed suitable for publication in PLOS ONE. Congratulations! Your manuscript is now with our production department. 

Kind regards, 

on behalf of

Dr. Alfio Spina 

Academic Editor

PLOS ONE